# Electrochemical Biosensor for Detection of the *CYP2C19*2* Allele Based on Exonuclease Ⅲ

**DOI:** 10.3390/mi14030541

**Published:** 2023-02-25

**Authors:** Siling Chen, Rongjun Yu, Ying Li, Jiangling Wu, Jingfu Qiu, Xinyi Huang, Jianjiang Xue

**Affiliations:** 1Department of Clinical Laboratory, University-Town Hospital of Chongqing Medical University, Chongqing 401331, China; 2School of Public Health and Management, Chongqing Medical University, Chongqing 400016, China; 3Department of Clinical Laboratory, First Affiliated Hospital of Guangxi University of Chinese Medicine, Nanning 530023, China

**Keywords:** clopidogrel, *CYP2C19*2* gene, electrochemical biosensor, exonuclease III, calixarene

## Abstract

Currently, the therapeutic effect of clopidogrel differs considerably among individuals and is thought to be closely related to the genetic polymorphism of *CYP2C19*. The *CYP2C19*2* gene can reduce the antiplatelet aggregation effect of clopidogrel, which increases the risk of major cardiovascular adverse events in patients. In this research, we report a new type of biosensor for the highly sensitive detection of the *CYP2C19*2* gene based on exonuclease III assisted electric signal amplification and the use of calixarene to enrich electrical signal substances. Specifically, under the best conditions, the logarithmic concentrations of the analytes have a good linear relationship with the peak current in the range of 0.01 fM to 100 pM and the detection limit is 13.49 aM. The results have also shown that this method has good selectivity, high sensitivity, and stability, etc., and will provide a very promising application for the detection of the *CYP2C19*2* gene and other biological molecules by replacing corresponding nucleic acid sequences.

## 1. Introduction

With the growth of social economy, lifestyle change and the aging population, the incidence rate and mortality of coronary heart disease are continuously increasing, and this will seriously affect human health. Dual antiplatelet therapy combined with a P2Y12 receptor inhibitor and aspirin is the main treatment used for patients with acute coronary syndrome (ACS) and percutaneous coronary intervention (PCI) [1]. Clopidogrel is currently the most widely used P2Y12 receptor inhibitor. Although the antiplatelet aggregation effect of clopidogrel is remarkable, many clinical researches have shown that the platelets of different individuals have varying reactions to clopidogrel [2]. It has been reported that the antiplatelet effect in a large number of patients treated with clopidogrel was weakened or even ineffective in ischemic events such as stent thrombosis and recurrent myocardial infarction and these may still occur in some patients, that is, platelet hyperreactivity (HTPR) in treatment [3,4]. Clopidogrel is a thiophene pyridine precursor drug, which needs to be metabolized into active products in the liver through the cytochrome P450 oxidase (CYP450 enzyme) system (*CYP2C19*, *CYP3A4*, *CYP2B6*, etc.) in order to play role in antiplatelet aggregation [5]. The *CYP2C19* gene is responsible for encoding the *CYP2C19* enzyme [6]. Genetic variation of the *CYP2C19* gene leads to individual differences in *CYP2C19* enzyme activity [7]. The *CYP2C19* gene has the following main alleles: **1*, **2*, **3*, and **17* [8]. The *CYP2C19*2* and **3* alleles can reduce the activity of the *CYP2C19* enzyme and thus reduce the production of active metabolites of clopidogrel, which are called loss of function (LOF) alleles [9]. Studies have found that the allele frequency of the *CYP2C19* variant is ethnically different. The *CYP2C19*2* gene is the major *CYP2C19* variant found in Asians and the incidence in the Chinese population is 25.7%~36.6% [10]. Therefore, effective identification of the *CYP2C19*2* genotype is helpful in guiding the individualized treatment of clopidogrel.

Currently, there are many methods that have been used for the detection of *CYP2C19* gene polymorphisms, such as DNA-sequencing [11], real-time quantitative polymerase chain reaction (qPCR) [12] and multiplex PCR point mutation screening technology [13]. However, all the above-mentioned methods still have many disadvantages, such as the need for expensive reagents, difficulty in operation, poor stability, and long detection cycles. Thus, there is a need to construct a simpler and more sensitive technique for the rapid detection of the *CYP2C19*2* gene. Electrochemical biosensors have attracted a substantial amount of attention due to their low cost, simplicity, sensitivity, need for little sample preparation, and short response time [14].

To construct an electrochemical biosensor with high sensitivity, there are two common methods that can be used to improve the analytical performance. One key strategy is to design a new signal amplification mode. Among different electrochemical signal amplification strategies, enzyme assisted technology has gained attention [15]. Various nucleases have been widely used to assist in the recirculation of target [16]. Exonuclease III (EXO III), a common DNA treating tool, possesses a high cleavage activity. Double stranded DNA with a blunt or recessed 3′ end is the best substrate of EXO III. It gradually removes the single nucleotides along the 3′—hydroxyl terminus to the 5′ end of the double stranded DNA [17], which has no cleavage function for dsDNA containing a protruded 3′ end or single stranded DNA (ssDNA) [18]. EXO III–assisted electrical signal amplification has been widely used to develop novel biosensing platforms for nucleic acid detection because there are no special requirements for the sequence of substrates [19]. Graphene has many excellent characteristics including high surface areas, excellent conductivity, high chemical activity, low cost, and holds great potential applications and prospects in many technical fields [20]. Calixarene has an adjustable cavity size with rich and varied conformations, so it demonstrates good selectivity when binding with guest molecules [21]. Calixarene exhibited high supramolecular recognition and enrichment capability, which can form stable host–guest complexes via hydrogen bonding, specific stacking, or electrostatic interactions with various guest molecules [22]. Because of the above advantages, calix[6]arene can enrich the electroactive substance methylene blue (MB) resulting in a sharp increase in the current signal [23]. It has been determined that calixarene and carbon materials can form complexes by π-π interaction and formation of hydrogen bonds [24,25], thus we can synthesize new composites with excellent properties of calixarene and graphene simultaneously. Gold metal nanoparticles (AuNPs) have been extensively used to improve the sensitivity of biosensor due to their good conductivity, chemical stability, strong adsorption capability, and large surface area [26]. However, due to Van der Waals forces and high surface energy, AuNPs are prone to irreversible aggregation in solution. GO material has excellent loading capacity for nanoparticles, and it is a good carrier for dispersing metal nanoparticles. The in situ generated, reduced graphene oxide (RGO) is negatively charged and exhibits high absorption capacity toward positively charged Au. RGO is a good reducing agent for the formation of Au/RGO nanocomposites. In this research, AuNPs were used to load many auxiliary probes based on macrocyclic host SCX6 and RGO, so that the biosensor has supramolecular recognition and enrichment capabilities for MB. It also has good conductivity, and a large specific surface area.

In this study, we designed a sandwich type biosensor based on EXO III and host–guest complexes (SCX6/RGO) to enrich electroactive material for *CYP2C19*2* gene detection. The use of EXO III facilitated the circulation of analytes. In addition, the residual chain of the hairpin capture probe (HC) on the modified electrode was combined with the labelled probe (LP). The auxiliary probe (AP) possessed two different complementary areas with LP sequences. Therefore, sequence specificity detection was facilitated by using HC and LP, and long cascades were generated through multiple hybridizations between AP and LP, which increased the electrical signal and thus improved the sensitivity of the biosensor.

## 2. Experimental

### 2.1. Reagents and Materials

Sulfonated [6] calixarene (SCX6), and 6-mercapto-1-hexanol (MCH) were purchased from Tokyo Chemical Industry Co., Ltd. (Tokyo, Japan). Graphite oxide (GO) was purchased from Nanjing XFNANO Materials Tech Co., Ltd. (Nanjing, China). EXO III was purchased from Thermo Fisher Scientific Co., Ltd. (Shanghai, China). Methylene blue, tris(2-carboxyethyl) phosphine hydrochloride (TCEP), and the gold chloride (HAuCl_4_∙4H_2_O) solution were obtained from Aladdin Industrial Corporation (Shanghai, China). Synthetic oligonucleotide sequences in this experiment were bought from Sangon Biotechnology Co. (Chongqing, China) and are displayed in Appendix A. Clinical serum samples were obtained from the University-Town Hospital of Chongqing Medical University (Chongqing, China). The solutions and buffers used in this experiment are shown in Appendix A.

### 2.2. Equipment

The electrochemical detections, such as DPV (Differential Pulse Voltammetry), EIS (Electrochemical Impedance Spectroscopy) and CV (Cyclic Voltammetry) were all performed on an AUTOLAB PGSTAT302N electrochemical workstation (Metrohm Technology Co., Ltd., Herisau, Switzerland). Transmission electron microscopy (TEM) images were obtained with a JEM 2100F microscope (JEOL, Tokyo, Japan). The structure of nanomaterials was characterized by an infrared spectrometer (FTIR) (Nicolet iS10, Thermo, Waltham, USA). The thermogravimetric analysis (TGA) results were recorded by a TGA instrument (NETZSCH STA 449 F3/F5, Selb, Germany). X-ray photoelectron spectroscopy (XPS) analysis results were obtained using a 250Xi photoelectron spectrometer (Escalab, Thermo, Waltham, MA, USA).

### 2.3. Preparation of Au/SCX6/RGO-MB Nanocomposites

The SCX6/RGO composite was prepared according to the previous method [27,28]. To obtain the nanocomposites, first, 10 mg GO materials were dissolved in 10 mL ultrapure water by ultrasonication for 1 h at room temperature(25 ± 1 °C) to obtain the yellow brown aqueous suspension. At the same time, 20 mg SCX6 was also processed in the above method. Then, 10 mL 2.0 mg/mL SCX6 solution was added to the 10 mL brown aqueous suspension and stirred at room temperature for over 12 h to obtain the mixed solution. With 1.0 M NaOH, the pH of the mixture solution was adjusted to pH 8–12. Afterwards, the mixture solution was transferred into a round bottom flask and refluxed at 90 °C in an oil bath for over 5 h. After naturally cooling down to room temperature (25 ± 1 °C), the black complex (SCX6/RGO) was obtained by repeated centrifugation, three times at 12,000 rpm for half an hour and washed with deionized water. The black complex was dissolved in 10 mL ultrapure water by ultrasonication for 1 h to obtain a black aqueous suspension. Next, 100 μL of 10 mM HAuCl_4_ solution was added slowly, dropwise to the black solution and magnetically stirred for 24 h at room temperature. After centrifuging three times (half an hour each time), 0.1 mg methylene blue powder was added to the above black suspension and stirred for 12 h. After that, the Au/SCX6/RGO-MB nanocomposites were successfully obtained.

### 2.4. Fabrication of the Proposed Biosensor

Before use, HC underwent denaturation at 95 °C for 5 min and then gradually cooled down to room temperature (25 ± 1 °C). Afterwards, the solution of hairpin structure DNA was stored at 4 °C. The bare glassy carbon electrode (GCE, Φ = 3 mm) was polished with 0.3 μm and 0.05 μm alumina powder dispersion sequentially, and then sonicated with ultrapure water, ethanol, and ultrapure water again for several minutes. To obtain the AuNPs layer, the bare electrode was submerged into 1% HAuCl_4_ solution for electrochemical deposition at the potential of −0.2 V for 60 s. Before incubation with the electrode, the HC was diluted to 1 μM with 10 mM TCEP buffer for 1 h at room temperature, which reduces the formation of disulfide bonds. Next, 10 μL of 1 μM HC was added onto the prepared Au/GCE electrode surface and then incubated for 16 h at 4 °C. The hairpin capture probe was conjugated to the AuNPs layer of the GCE electrode through Au-S bonding [29]. The electrode surface was rinsed with washing buffer to remove the unbound HC. Next, 10 μL 1 μM MCH solution was added to the electrode and incubated for 1 h to prevent non-specific adsorption to generate a well-aligned DNA monolayer. After the modified electrode was rinsed, 10 μL of 1 μM target DNA (tDNA) was added and bioconjugated for 2 h. Then, 10 U EXO III was used to remove the single nucleotides of dsDNA from the 3′ to 5′ end at 37 °C for 50 min. After that, the pre-treated electrode was incubated with Buffer1 (80 μL mg/mL Au/SCX6/RGO-MB and 10 μL 1 μM LP incubated at 4 °C for 12 h in advance) and 10 μL 1 μM AP for 2 h at 37 °C after which EXO III was washed with washing buffer thoroughly. Finally, the developed biosensor was washed with washing buffer and prepared for the further experiments.

### 2.5. Electrochemical Detection

All electrochemical experiments were completed in a conventional three-electrode system, which consisted of a working electrode (GCE electrode), a reference electrode (saturated calomel electrode), and an auxiliary electrode (platinum wire). DPV was performed in 0.1M PBS (phosphate buffered saline, pH 7.4). Voltages between −0.6 V and 0.2 V, modulation amplitudes of 0.07 V, modulation times of 0.05 s, and interval time of 0.2 s were used to conduct the DPV measurements. The EIS parameters included a 10 mV amplitude and a frequency sweep range from 10^−1^ to 10^5^ Hz. With a scan rate of 100 mV s^−1^, the CV measurements were conducted by scanning potential between −0.2 and 0.6 V. The above measurements were conducted at room temperature (25 ± 1 °C).

## 3. Results and Discussion

### 3.1. Principle of the Electrochemical Biosensor

The principle of the developed biosensor for *CYP2C19*2* gene detection is shown in Figure 1. The biosensing process of the developed biosensor consisted of two parts: EXO III assisted the target DNA recycling and forms a sandwich type electrochemical biosensor based on host–guest complexes. In this method, the HC was modified by a mercapto group and possessed a complementary area to target DNA. Firstly, the bare GCE electrode was submerged into the HAuCl_4_ solution for electrochemical deposition to create the AuNP layer. Then, the HC was immobilized on the modified electrode via Au-S bonding. Next, the target DNA was added to the reaction to open the stem-loop of HC and form double stranded DNA with a blunt 3′ end. After that, the target DNA was released because of Exo III being used to remove the single nucleotides of the dsDNA from 3′ to 5′ terminal. The released target DNA can be used for the next reaction cycle thus promoting the circulation of analytes. After the above reaction, the residual chain of HC on the electrode was combined with the Au/SCX6/RGO-MB-LP (Au/SCX6/RGO-MB and LP incubated at 4 °C for 12 h) bioconjugate via Au-S bonding. The AP possessed complementary regions with two different areas of LP. Long cascades were generated through multiple hybridization between AP and LP. Finally, the sandwich type electrochemical biosensor based on host–guest complexes was formed. The host–guest complexes can enrich the content of electrochemical mediators of methylene blue, leading to current signal increases apparently. The electrochemical signals (reduction peak generated by MB) were detected by DPV. Without target DNA, HC sufficiently maintained the stable hairpin structure, the subsequent complex was not immobilized on the modified electrode, and there was no obvious electrochemical signal change.

### 3.2. Characterization of Electrode Assembly Process

In this study, we used electrochemical devices to characterize the interface properties through the EIS method, which investigated the step-by-step preparation of the biosensor. According to the Nyquist diagram, the curve’s semicircle diameter equals the charge transfer resistance(Ret) [30]. The EIS curves of stepwise modified electrodes are shown in Figure 1A. The results of impedance spectroscopy indicated that the resistance of the GCE electrode after gold electrodeposition (curve b with a value of about 373 Ω and its confidence interval (CI) is (195, 453)), was lower than that of the bare GCE electrode (curve a) value of about 756 Ω, CI (579, 818), indicating that AuNPs were successfully attached to the electrode surface. It showed that AuNPs had good conductivity and can promote electron transfer. Due to the electrostatic repulsion between the negatively charged oligonucleotide phosphate skeleton and [Fe (CN)_6_]^3−/4−^ in solution, the Ret increased (curve c with a value of about 2354 Ω, CI (1968, 2532)) when HC was immobilized on the modified Au/GCE electrode by Au-S bonding, showing that the immobilization of the HC was successful. After the target DNA combined with the HC on the electrode surface, the Ret increased appropriately (curve d), the value was about 2913 Ω, CI (2858, 3131). Both electrostatic repulsion and the steric hindrance of organic molecules severally hindered charge transfer on the interface. After the reaction of EXO Ⅲ, the Ret decreased apparently (curve e), the value was about 1467 Ω, CI (1468, 1712). In this case, target DNA was hybridized with HC and formed ds DNA. The dsDNA was partially digested with the activation of the cleavage reaction. However, when the probe nanocomposite (Au/SCX6/RGO-MB-LP-AP) was hybridized with the residual HC strands on the electrode surface, inapparently changed Ret (curve f) was obtained and the Ret value was about 1309 Ω (CI (1242, 1442)) due to the high conductivity of AuNPs and RGO. At the same time, it can be observed that when only LP, AP and MB (without Au/SCX6/RGO nanocomposite) were added to be hybridized with residual HC strands on the electrode surface compared with curve e, the semicircle diameter significantly increased (curve g) and the Ret value rose to about 4086 Ω, CI (3832, 4221). This was owing to the obvious steric hindrance and electrostatic repulsion of numerous LP and AP hybridizations. It was also a satisfactory proof of the high conductivity of the nanocomposite. The findings from CV measurements (Figure 1B) were in excellent agreement with the Nyquist diagrams, which showed that the redox peak changed according to the gradual modification process. Both the Nyquist diagrams and CV results indicated that the proposed biosensing platform was successfully constructed as shown in the schematic diagram.

### 3.3. Characterization of Nanocomposites

To characterize the particles of the Au/SCX6/RGO nanocomposites, the morphology and microstructure were observed using a field emission transmission electron microscope and we found that AuNPs (about 5 nm) were evenly distributed on the surface of SCX6/RGO (a single-layer sheet) (Figure 2A). The RGO and SCX6/RGO were prepared for thermogravimetric characterization and the analytical results are shown in Figure 1B. We found that the lost mass of RGO and SCX6/RGO at 600 °C was approximately 62.6% and 41.8%, respectively. Therefore, the lost mass resulting from the decomposition of SCX6 was 21.8%, showing that SCX6/RGO nanocomposite was successfully synthesized.

As shown in Figure 2C, the FTIR spectrum demonstrated the presence of −OH (3440 cm^−1^), C–O/C–C (1046 cm^−1^), and C=C (1631 cm^−1^) compounds in the RGO material. We also found that the SCX6/RGO contained the above characteristic peak of RGO. In addition, it was observed that the characteristic peaks −SO_3_− (1160 cm^−1^) of SCX6 also appeared on the SCX6/RGO composite, indicating that the SCX6/RGO composite was successfully prepared.

Figure 2D demonstrates the XPS patterns of Au/SCX6/RGO and proves the existence of C, O and Au in the complex. Figure 2E illustrates the RGO C1s spectrum, revealing five components containing carbon bonds: C–C/C=C (284.81 eV), C–O (286.13 eV), C=O (286.96 eV) and O–C=O(288.76 eV). In addition, compared with the C1s spectrum of Au/SCX6/RGO in Figure 2F, we found that C–C/C=C was the main functional component in Au/SCX6/RGO. All the above results showed that the Au/SCX6/RGO was successfully synthesized.

### 3.4. Optimization of Experimental Conditions

In this study, some experimental parameters were optimized to obtain the best analytical performance of the fabricated biosensor. As shown in Figure 3A, the peak currents increased gradually as the DNA strand incubation time increased from 30 min to 2 h. The peak current reached a stable plateau about −9.56 μA at 2 h, indicating that the incubation between DNA strands increased the MB current dramatically, while the peak currents changed slightly over 2 h. According to the current response, 2 h was employed as the optimal incubation time. The peak currents were dependent on residual HC strands which were not only correlated with the incubation time to form the double stranded DNA, but also related to the cleavage reaction of EXO III. Considering the high cost, 1 U/μL EXO III was used in the assay, and its cleavage time was optimized. As shown in Figure 3B, the peak currents were positively correlated with the reaction time of EXO III. The peak currents increased gradually within 50 min and then kept steady status. The maximum electrical signal of −10.06 μA was observed when the cleavage time reached 50 min. Therefore, 50 min was used as the optimal EXO Ⅲ incubation time. In addition, the amount of hairpin capture probe also plays a key role in the biosensor. With the increase in HC concentration (ranging from 0.2 to 1.2 μM), the peak currents of the biosensor increased as demonstrated in Figure 3C. When the concentration of HC reached 1 μM, the peak currents decreased slightly, which indicated that the high concentration of HC reduced the hybridization efficiency of nucleic acids due to the steric hindrance effect between nucleic acids. Therefore, the optimal reaction concentration of HC was 1 μM.

### 3.5. Analytical Performance of the Developed Biosensor

Under the optimal experimental conditions, the detection of different concentrations of target DNA were investigated by DPV measurements to obtain the best analytical performance of the constructed biosensing platform. The response signals upon target DNA concentrations in the 0–100 pM range are shown in Figure 3A. According to Figure 3A, the DPV peaks were increased with the increase in *CYP2C19*2* gene concentration. In the concentration range of 0.01 fM to 100 pM, the linear relationship between the peak current and the logarithm concentrations of the target DNA was plotted (Figure 4B). The regression equation obtained from this curve was Y = −1.31logC − 27.67, R^2^ = 0.99 (C is the *CYP2C19*2* gene concentration and R^2^ is regression coefficient). The calculated limit of detection (LOD) was obtained as low as 13.49 aM (S/N = 3), S and N represent the standard deviation of the blank and the slope of the calibration plot, respectively. The comparison between this biosensor and other existing methods for *CYP2C9*2* gene detection are shown in Appendix A, demonstrating that the biosensor established in this study has the lowest detection limit. As a result, this biosensor could be used for *CYP2C9*2* gene detection with excellent electrochemical properties.

### 3.6. Specificity and Stability of the Biosensor

To verify the selectivity of the constructed biosensor, the DPV current responses generated from the same concentration of interfering substances with different mismatched bases were recorded to compare with target DNA. Under the same experimental conditions, 1 fM target DNA was replaced with single-base mismatch target(1-MT), double-base mismatch target(2-MT), non-complementary DNA(NC) and blank solution (each 1 pM), respectively. The results are displayed in Figure 5A. We found that the blank current peak was about −3.24 µA due to the electrostatic adsorption between nucleic acid skeleton and methylene blue. The current of 1-MT, 2-MT and NC were almost identical to blank, even though the target DNA concentration was lower by a factor of 100 times than the concentrations of the interfering substances. The peak currents increased considerably only when the fabricated biosensor was incubated in target DNA, indicating that the designed biosensor had good selectivity. It can effectively discriminate interference with different mismatched bases and could meet the experimental requirements. Under the optimized conditions, the stability of the fabricated biosensor was also verified. Prior to use, the fabricated biosensor was stored at 4 °C and the DPV signals were measured one week later. Figure 5B indicates the comparison of the signals of the five groups of biosensors before and after they were stored for one week. We found that the electrochemical signals were only slightly reduced, the current signal remained at 86.81–92.94% compared with the initial signals, demonstrating that this designed biosensor was coupled with acceptable stability.

### 3.7. Practical Applications of the Biosensor in Real Samples

To further study the applicability of the proposed biosensor in clinical applications, we added five groups of analytes with different concentrations (0 fM, 0.1 fM, 1 fM, 10 fM, 50 fM) to the 10-fold diluted human serum samples, and signals of the obtained samples were subsequently recorded by DPV measurements. The recovery calculated by the standard addition method was found to fluctuate from 96.67% to 107.98% and the relative standard deviations (RSD) varied from 0.39% to 2.70%. The recovery and RSD are all within an acceptable range. The obtained results demonstrated that the biosensor had potential applicability in clinical applications. The detection results of the *CYP2C19*2* gene in human serum samples are summarized in Appendix A.

## 4. Conclusions

In this research, we reported a new type of biosensing platform for highly sensitive detection of the *CYP2C19*2* gene. Compared to other *CYP2C19*2* gene sensing methods, this sensing platform possesses some fine characteristics: first, EXO Ⅲ assisted target DNA recycling was used to design a sensitive biosensor due to its easy design, high amplification efficiency and no specific requirements for substrate sequence. Secondly, the residual chain of HC on the electrode was combined with AL and LP. Long cascades were generated through multiple hybridizations between AP and LP. Finally, the host–guest complexes (Au/SCX6/RGO) can enrich the electrochemical mediators of MB, which lead to the current increasing dramatically. The fabricated biosensor displayed wide linear ranges between peak currents and the logarithm of *CYP2C19*2* gene concentrations (0.01 fM to 100 pM), achieved an ultra-low detection limit (13.49 aM). Furthermore, the biosensor showed good selectivity with interfering substances in human serum, making it a potential method for clinical application.

## Data Availability

All data generated or analyzed during this study are included in this article and the Additional Information.

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
