# Peer review of "Electrochemical Biosensor for Detection of the CYP2C19*2 Allele Based on Exonuclease Ⅲ"

_micromachines, 2023, doi:10.3390/mi14030541_

Round 1
Reviewer 1 Report
The ms reports a study on the development of an electrochemical biosensor for ultrasensitive detection of the 2CYP2C19*2 allele based on exonuclease Ⅲ.
Overall, the ms is interesting however revision is needed.
1) fig 1 should report the baseline. why the capacitive current is so high (e.g. CVs are tilted?). The x-axis should quote the potential vs Ag/AgCl or other reference electrode?
2) what's happen above 10-11 M? see Fig. 4B
3) I don't see a great selectivity from Fig. 5. Why this?
4) how is it possible a limit of detection as 0.00104 fM? that would mean a precision of 10-20M. I don't trust that. How this was calculated?
Reviewer 2 Report
The manuscript entitled „Electrochemical biosensor for ultrasensitive detection of the CYP2C19*2 allele based on exonuclease Ⅲ” has in attention a biosensor based on exonuclease III for sensitive, selective and stable detection of CYP2C19*2 allele.
The authors are kindly requested to consider the following recommendations for the readers' convenience:
- express the detection limit as attomolar;
- page 2, lines 63-64, rephrase ”...electrochemical signal amplification technologies, enzyme assisted signal amplification technology...” to avoid repetition;
- extend the discussion on the biosensor characteristics (sensitivity, linearity, reproducibility, shelf life) in the Results section;
- insert the error bars in all figures;
- define all used abbreviations (DPV, EIS, etc.);
- Correct (uniformize) the degree Celsius in the entire manuscript;
- Mention how long was centrifugation performed;
- line 256 - provide more details for the ”As a result, 2 h was employed as the optimal experimental condition.”
- give more details on the experiments performed for the interferents.
Reviewer 3 Report
The manuscript ID micromachines-2175322 mainly presents a study about a particular electrochemical biosensor for detection of the CYP2C19*2 gene and other biological molecules by replacing corresponding nucleic acid sequences. Please see below a list of comments to the authors:
- The parameters for the preparation method of the hybrid nanostructures should be better described in order to see that the report is systematic.
- Please comment about the influence of the size of the nanoparticles in the results.
- From the micrograph seems that the nanoparticles are anisotropic. How is the influence of the anisotropic shape of the nanoparticles in the electrochemical measurements?
- It should be noted that the integration of Au and RGO drastically changes the electrical and electromagnetic response of the system and the assistance of light can be considered in a perspective for future work. The authors are invited to discuss in this respect and see for instance: https://doi.org/10.1364/OE.27.007330
- It is not clear how were selected the Au nanoparticles in combination with reduced graphene oxide in respect to other comparative materials for this work. You can see for instance: https://doi.org/10.1039/D2AY01592G
- Please report how is the reproducibility of the samples, and how was determined the volume fraction of the Au nanoparticles decorating the reduced oxide graphene?
- How was proposed the concentration of the nanocomposites that should be employed to perform the biosensing? In this respect the analysis of an experimental calibration curve to guarantee the sensibility would be welcome.
- In there any plasmonic contribution from the nanocomposites in the biosensing process? Please argue.
- In order to better justify the selection of the citations and improve the presentation of the topic, it is suggested to split the collective citation form in individual form.
- The concentration range and detection limit together to advantages and disadvantages of the different representative methods for biosensing could be compared and summarized in a table.
Reviewer 4 Report
In this manuscript, the DNA sensors based on graphene oxide – sulfonated calix[6]arene composite has been developed for sensitive determination of the CYP2C19*2 gene sequence. Although the amplification principle is not new and has numerous applications, the use of host-guest complexation resulted in sufficient increase of the signal. For this reason, the manuscript can be recommended for publication after some changes. First and foremost, the announced accuracy parameters are unreachable and can be attributed only to some mistakes in the measurement protocols. I believe that the authors just repeated the DPV with the same layer and methylene blue additive. If you take bare GCE and add methylene blue, the deviation of the signal for three individual electrodes (and three separate additives of methylene blue) will differ for much more than 0.39%. Second, it is not clear, what signal (reduction, oxidation) was measured for methylene blue. Third, the description of EIS data is very simplified and naïve. Instead, the equivalent circuit choice should be given together with metrological assessment of the results collected in the Table.
Other comments:
Title: it is recommended to avoid emotional meaning like “ultrasensitive”. This term also contradicts with the assessment “good” given in the Abstract for detection performance (what’s that?) and selectivity.
Abstract: The sentence “We believe that we can guide clinical medication by screening sensitive genotypes and select an appropriate antiplatelet regimen to improve the prognosis of coronary heart disease” seems exhausting and can be removed.
Abstract: semi-logarithmic scale of calibration should be mentioned.
Introduction: Line 73 “Graphene oxide has … excellent conductivity” – graphene has, graphene oxide has not, please correct.
Citation [22] is wrong. First, it describes calixcarbazole, not calixarene, second , this is a particular research but not the description of common properties of calixarenes – please find a review instead.
Line 80: “calix[6]arene (SCX6)” – this acronym was used for SULFONATED calix[6]arene, please correct
Line 83 “hydrogen interaction” – probably, formation of hydrogen bonds would sound better
Lines 88-89 “the biosensor has excellent 88 supramolecular recognition, enrichment capabilities, superior conductivity, and large specific surface area.” – please re-write the sentence. Some of the features can be attributed only to the signal, not to biosensors, others to the transducer. And try to avoid the terms “superior”, “excellent” etc. – give the reader chance to have own meaning about biosensor characteristics.
Experimental: Line 104 - GO acronym has been already introduced in Line 73, MB in Line 81, - please avoid duplication
Line 116 – “s DPV, EIS and CV” – explained the acronyms please
Section 2.3 and through the text – “ml” should be substituted with “mL”
Lines 136, 147, 182 – HAuCl4 – check subscript
Line 166 – “3-mm-diameter GCE electrode” – the information is duplicated (see Section 2.4, line 144)
Line 168 PBS – give explanation of the acronym. Be sure that this acronym is used for phosphate buffer saline containing NaCl. Otherwise, use PB acronym.
Line 172 – check superscript (100 mV s-1)
Lines 172-173 “by scanning by scanning” – remove duplication, add the word ‘potential’
Section 3.1 – title is inappropriate, mechanism can be attributed to the biosensor signal, not to biosensor as such
Line 178 – Scheme 1 is placed very far from first mentioning in the text
Line 179, 193 “super sandwich type electrochemical biosensor” – please explain what the term super means in this context?
Line 194 “can enrich the electrochemical mediators MB,” – enrich what? Content, concentration? – please add
Line 201 – “EIS technology” – method, not technology
Line 202 “EIS curve's semicircle” – this is the description of the Nyquist diagram, not EIS in general, ref, [33] should be changed by general description of the EIS theory, not particular research
Figure 1 and its description – these are not EIS cures but the Nyquist diagrams, please change.
Section 3.2 – please give the equivalent circuit used for fitting, specify the equilibrium potential and provide appropriate Ret values with confidence intervals instead of quantitative description of the results.
Line 206 “AuNPs was successfully attached” – WERE successfully attached
Line 209 – [Fe(CN)6] – check subscript
Line 224 – Figure 1(B) – please add scan rate to the caption.
Lines 239, 241 – cm-1 – check superscript
Line241 -SO3- - check subscript
Line 254 “stable platform” – plateau, not platform
Section 3.5 – please explain the conditions of parallel measurements – individual sensors or three records of DPV? Second case is inappropriate, first experiments would give much higher deviation for sure.
Line 276 “Y=-1.2675logC-27.00357, R2 = 0.99194” – please reduce the number of decimals, add dimension of concentration, add the current in the blank experiment with no analyte, R2 – check superscript
Line 333 (Figure caption) – please remove last sentences (“This is a table. Tables should be placed in the main text near to the first time they are cited”)
Line 303 “five groups of analytes” – here is only one analyte, others are called as interferences.
Lines 306 – please take into consideration that percentage should have lower number of significant digits against absolute values. The RSD (0.39%) is impossible for electrochemical measurements on solid electrodes even for bare GCE – please give the protocol of metrological assessment. One electrode or three DPV runs????
Round 2
Reviewer 1 Report
I'm not happy about the revision performed by the authors.
Point 1: when i asked to insert the baseline, I meant the CV of GCE without redox mediator;
Point 2: authors should add also points above 10-11M to see when the plot plateaus. The explanation given is nonsense because it is equivalent to data omission;
Point 4: a LoD of 10-19 is again not credible considering that the background current seems very high.
From what I can see from Fig. 5 the system is not selective as the signal seems very much affected by interferences.
Reviewer 2 Report
The authors have adequately addressed all the reviewers' comments.
Author Response
We are very thankful to the reviewer for the careful reading of our manuscript. We appreciate the helpful and constructive comments of the reviewer.
Reviewer 3 Report
Some of the points raised in the review stage have been addressed; however fundamental issues related to the selection of the size and shape of the nanoparticles, together to the influence of the integration of Au and RGO for this particular sensing performance are still missing. In my opinion, information about these aspects should be analyzed, if possible experimentally, or critically discussed in order to improve this work to be a base for future research.
Reviewer 4 Report
The authors mostly followed the Reviewer's comments. However, some problems remained:
1. Line 196 - indeed, REDUCTION currents are recorded, not oxidation (see FIg.4)
2. EIS datashould be provided with confidence intervals of Ret values
3. Line 283: please reduce the number of decimals (Y=-1.268 logC-27.00)
Round 3
Reviewer 1 Report
See comments to Editors. I put reconsider after major revision, but I rather prefer to abstain on this work. I suggest the Editor to make a final decision.
Reviewer 3 Report
I agree with the points clarified by the authors. In my opinion, the manuscript can be considered for publication in present form.
Author Response

(The authors gave the same response as above.)
